# A family-based intervention for prevention and self-management of disabilities due to leprosy, podoconiosis and lymphatic filariasis in Ethiopia: A proof of concept study

**Anna T. van't Noordende**[1,2,3]*, **Moges Wubie Aycheh**[4], **Tesfaye Tadesse**[5], **Tanny Hagens**[6], **Eva Haverkort**[1], **Alice P. Schippers**[1,7]

**1** Disability Studies in the Netherlands, Almere, the Netherlands, **2** NLR, Amsterdam, The Netherlands, **3** Erasmus MC, University Medical Center Rotterdam, Rotterdam, The Netherlands, **4** Debre Markos University, Debre Markos, Ethiopia, **5** Ethiopian National Association of Persons Affected by Leprosy (ENAPAL), Addis Ababa, Ethiopia, **6** The Leprosy Mission Ethiopia, Addis Ababa, Ethiopia, **7** Amsterdam University Medical Centre, Location VU Medical Center, Amsterdam, The Netherlands

* a.vt.noordende@nlrinternational.org

**Data Availability Statement:** All data are available from the infoNTD website and can be downloaded

## Abstract

A key issue for persons with leprosy-, lymphatic filariasis- and podoconiosis-related disabilities is the life-long need to practice self-management routines. This is difficult to sustain without regular encouragement and support of others. Family-based support may be a sustainable and feasible strategy to practice self-management routines. This proof of concept study aimed to develop and pilot a family-based intervention to support prevention and self-management of leprosy, lymphatic filariasis and podoconiosis-related disabilities in Ethiopia.

We used a quasi-experimental pre/post intervention study design with a mixed methods approach. The study population included persons affected by leprosy, lymphatic filariasis and podoconiosis and their family members. All persons affected had visible impairments due to their condition. We collected physical impairment outcomes, data on activity limitations, stigma and family quality of life using the SALSA scale (range 0–80), the SARI stigma scale (range 0–63) and the Beach Centre Family Quality of Life scale (range 0–125) and conducted in-depth interviews and focus group discussions. Quantitative data were analysed using paired t-tests, unequal variances t-tests, linear regression and binary logistic regression. Qualitative data were coded using open, inductive coding and content analysis.

The family-based intervention consisted of self-management of disabilities, awareness raising and socio-economic empowerment. The intervention was delivered over several monthly group meetings over the course of several months. A total of 275 (100%) persons affected attended at least one session with a family member, and 215 (78%) attended at least three sessions. There was no significant improvement in eye and hand problems after the intervention. However, foot and leg impairments, number of acute attacks, lymphedema and shoe wearing all significantly improved at follow-up. In addition, family quality of life significantly improved from 67.4 at baseline to 89.9 at follow-up for family members and from

via https://www.infontd.org/sites/default/files/2021-01/fba_ethiopia_routine_data.txt and https://www.infontd.org/sites/default/files/2021-01/fba_ethiopia_-_quantitative_data_baseline_follow-up.txt.

**Funding:** This study is funded by the Leprosy Research Initiative Foundation (LRI, leprosyresearch.org) under project number 705.17.30. The funders had no role in study design, data collection and analysis, decision to publish, or preparation of the manuscript.

**Competing interests:** The authors have declared that no competing interests exist.

76.9 to 84.1 for persons affected (p<0.001). Stigma levels significantly decreased from 24.0 at baseline to 16.7 at follow-up (p<0.001). Activity levels improved, but not significantly.

This proof of concept study showed that the family-based intervention had a positive effect on impairments and self-management of disabilities, family quality of life and stigma. We recommend a large-scale efficacy trial, using a randomised controlled trial and validated measurement tools, to determine its effectiveness and long-term sustainability.

## Author summary

A key issue for persons with leprosy-, lymphatic filariasis- and podoconiosis-related disabilities is the life-long need to practice self-management routines. Family-based support may be a sustainable and feasible strategy to practice self-management. This proof of concept study aimed to develop and pilot a family-based intervention to support prevention and self-management of leprosy, lymphatic filariasis and podoconiosis-related disabilities in Ethiopia. We collected data on physical impairments, activity limitations, stigma and family quality of life, and conducted in-depth interviews and focus group discussions. The family-based intervention consisted of self-management of disabilities, awareness raising and socio-economic empowerment. The intervention was delivered over several monthly group meetings. A total of 275 persons affected attended at least one session with a family member. There was no significant improvement in eye and hand problems after the intervention. However, foot and leg impairments, number of acute attacks, lymphedema and shoe wearing all significantly improved. In addition, family quality of life significantly improved, and stigma levels significantly decreased after the intervention. Activity levels improved, but not significantly. This proof of concept study showed that the family-based intervention had a positive effect on impairments and self-management of disabilities, family quality of life and stigma.

## Introduction

Leprosy, lymphatic filariasis (LF) and podoconiosis are three skin-related Neglected Tropical Diseases (NTDs) [1]. All three conditions exhibit cutaneous manifestations such as patches and ulcers (leprosy), swollen limbs (LF and podoconiosis) and wounds, nodules or swelling (all three) [2–5]. If not diagnosed and treated early, all three conditions can lead to temporary and permanent impairments [1,2,6].

Leprosy, LF and podoconiosis-related impairments are major determinants of stigma and participation restrictions [7–9]. Stigma and physical impairments may also hamper people's daily functioning, for example their ability to work. This may deteriorate the economic situation of persons affected and may impose a social and economic burden on already marginalized families [5,10–13]. This while most impairments, particularly visible impairments such as wounds, swelling and contractures, are largely preventable. Relatively simple methods exist for self-management of impairments that can be practiced at home, without the need for a lot of medical supplies. Many of these methods for prevention and self-management of disabilities are suitable for use across different skin-related NTDs [14,15]. Too often, however, these methods are not taught to patients with neuropathic limbs or lymphedema, or if taught, they are not consistently practiced. Good self-care management practices are crucial to prevent

further impairments, reduce symptoms, preserve quality of life and improve the ability to participate in work and social activities [16,17].

A key issue for persons with leprosy-, LF- and podoconiosis-related disabilities is the life-long need to practice such self-management routines. This is difficult to sustain without regular encouragement and support of others. A strategy shown to be successful is the formation of self-care groups in which persons affected by leprosy, LF and podoconiosis-related impairments support each other [18–20]. Self-care groups however, often have limited long-term sustainability and members may have problems in accessing the groups, for example because of financial or geographical barriers [19]. Family-based support may be a more sustainable and feasible strategy to practice self-management routines.

Several studies have indicated that family support is a highly significant factor in adherence to self-care [21,22]. When family functioning is not optimal, it is difficult to manage self-care and self-care is not as effective as it could be [21,23]. Strengthening social support and quality relationships, as is done through a family-based intervention, may also improve mental wellbeing of persons affected and their family members [24,25]. People with adequate social support seem to cope and adjust better with stressful events [26]. Since family-based support is practiced at home, no travel is required, and practicing self-care can be done at more flexible hours–a group facilitators is not required. Even though family-based support seems a sustainable and feasible strategy to practice self-management routines, especially in areas with limited health resources, it has received little attention to date. To our knowledge, no family-based intervention for leprosy, LF and podoconiosis-related disability management exists to date.

This study aimed to develop and pilot a family-based approach to support prevention and self-management of leprosy, lymphatic filariasis and podoconiosis-related disabilities in the Ethiopian context. We hypothesized that the family-based intervention would also impact psychosocial outcomes such as (family) quality of life, stigma and activity levels. The ultimate aim of this study is to improve the lives of the families involved in the study. This study builds on results of a recently published study that explored the quality of life of families with a family member affected by leprosy, LF and podoconiosis [13].

## Methods

### Ethics statement

Ethical approval was obtained from the Debre Markos University Health Science College Research Review Committee, approval number: 199/16/09. In addition, the Awi zone (district) Health Desk office granted permission to conduct the study in the woredas. Since the literacy rate was low in our study area, all participants were verbally informed about the nature and objective of the study, of confidentiality of the data and the voluntary nature of the study prior to data collection. Verbal consent from each participant was obtained prior to data collection.

### Intervention development

The family-based intervention was developed by the research team over the course of a year, based on an exploratory study conducted in 2017 [13]. The exploratory study consisted of in-depth interviews and focus group discussions (FGDs) and included a total of 86 participants, persons affected and their family members. Participants were asked about their self-care practices, family quality of life, and about ideas for family-based interventions to support prevention and self-management of disabilities. We found that many of the problems reported in the exploratory study were not only related to physical impairments, but also caused by stigma and poverty [13]. It was therefore decided to include the following two components in the family-based intervention, besides self-management of disabilities: (1) awareness raising, and (2)

socio-economic empowerment. The family-based intervention was delivered over several monthly group meetings over the course of several months.

The main component of the family-based intervention, self-management of disabilities, used approaches that are appropriate for all three conditions as much as possible. These include inspection, foot hygiene using soap and water, skin care with removal of callous, application of ointment, elevation, exercises, bandaging and advice on appropriate footwear. The project made use of existing initiatives as much as possible, such as the WHO's Integrated morbidity management for LF and podoconiosis [27], the Ethiopian Ministry of Health's LF and podoconiosis morbidity management and disability prevention guidelines and the International Federation of Anti-Leprosy Association's guideline for prevention of disabilities in leprosy [28]. All participants received basic tools to practice self-care (Vaseline, a bucket, soap, and bandages if necessary). A detailed description of the family-based intervention can be found as supplementary information file (S1 Text).

## Study design and study site

We used a quasi-experimental pre/post intervention study design with a mixed methods approach. The study was conducted in the Awi zone, located approximately 470 kilometres Northwest of the capital of Ethiopia, Addis Ababa. The Awi zone is one of the eleven zones in the Amhara region. The Awi zone was selected because the area is endemic for leprosy, LF and podoconiosis [29–31]. The study was conducted in Zigem, Guagusa Shikudad (Injibara town) and Fagita lekoma (Addis Kidam town) woreda (district).

## Study population and sample

The study population included persons affected by leprosy, persons affected by LF and persons affected by podoconiosis ("persons affected") and their family members ("family members") living in the area where the intervention was offered. Since self-care practices for LF and podoconiosis are essentially the same, we did not make a distinction in our data between persons affected by LF and persons affected by podoconiosis.

This was a proof of concept study to see whether the family-based intervention had a positive impact on self-management and prevention of disabilities and to explore whether the intervention was feasible and acceptable. A sample size of 20–25 is adequate for studies that aim to demonstrate intervention efficacy and a sample size of 10–20 participants per group is adequate when trying to determine group differences [32]. To account for loss to follow up, we initially aimed to include at least 60 families in the intervention: 30 families of persons affected by leprosy and 30 families of persons affected by LF or podoconiosis. However, because we anticipated that participants would benefit from the intervention, we decided to include as many participants as possible, so more families would benefit.

We also administered questionnaires and conducted interviews and focus group discussions pre- and post-intervention. Results of the baseline interviews have been published previously and are therefore not reported in this paper [13]. We aimed to collect data until data saturation was reached. The participants in the qualitative sample will be a subset of those in the quantitative sample.

## Eligibility criteria

Because our intervention aimed to improve prevention and self-management of disabilities, all persons affected had to have visible impairments due to their condition. In addition, all participants had to live in one of the three districts in which the intervention was offered. Family members had to live in the same household as their affected family member and had to know

about the condition of their family member. Persons unable to give informed consent and persons younger than 15 years of age were excluded.

## Sampling methods

Participants were selected using convenience sampling. For persons affected by podoconiosis and LF, local health posts were visited and a list with eligible persons in the study area was prepared. Persons on the list were visited in their home and asked to participate in the family-based intervention. Persons affected by leprosy were contacted through organisations of persons affected by leprosy in the study areas. This strategy was chosen because of the close connection of the Ethiopian National Association of Person Affected by Leprosy (ENAPAL), one of the organisations involved in the implementation of the intervention, with organisations of persons affected by leprosy in the study areas. Persons affected selected one family member from among those living in the same household, to participate in the intervention. This was done based on their availability.

## Data collection

The baseline study was conducted in 2017 and 2018, the intervention ran from February to October 2019 and the follow-up study was conducted in October and November 2019. Mixed methods were used to get an understanding of the impact of the family-based intervention. Physical impairment outcomes were collected, three questionnaires were used, and in-depth interviews and focus group discussions were conducted. All data methods were administered pre- and post-intervention.

Physical impairment outcomes were assessed by noting down if impairments were present for eyes, hands and feet for leprosy (we scored impairments as either not present/0 or present/1), and frequency of acute attacks, shoe wearing behaviour and foot and leg circumference for LF and podoconiosis. Eye problems included difficulty seeing at six metres distance and lagophthalmos. Leg and foot problems included swelling, foot drop and shortening or loss of toes. Hand problems included claw hand and shortening or loss of fingers. These data were routinely collected from all persons affected during each family-based intervention group meeting.

The 25-item Beach Centre Family Quality of Life (FQoL) scale (range 0–125, with higher scores indicating higher family quality of life) was used to assess family quality of life of persons affected and their family members. The scale contains five subscales: family interaction, parenting, emotional well-being, physical/material well-being, and disability-related support [33]. The FQoL scale has not been validated in Amharic. The FQoL scale was translated from English to Amharic, the translation was checked by translating the instrument back into English again using different interpreters. The FQoL scale was piloted tested among 20 participants before use.

The 20-item Screening of Activity Limitation and Safety Awareness (SALSA) scale was used to assess activity limitations of persons affected (range 0–80, with higher scores indicating more activity limitations). The questionnaire was developed in five countries in four continents among people affected by leprosy and diabetes [34]. The SALSA scale has been found to be a valid instrument to measure activity limitations in persons with a locomotor disability also [35]. In order to be able to compare results between conditions, we decided to use the scale for all three conditions. The SALSA scale has been validated in Amharic [36].

The 21-item SARI Stigma Scale (SSS) was used to assess stigma (range 0–63, with higher scores indicating more stigma experience). The SSS has been developed to assess leprosy-related stigma and assesses four aspects of stigma: personalised (experienced) stigma,

disclosure concerns, internalised stigma and anticipated or perceived stigma. We believe the SSS can be used to assess stigma experience in other NTDs also, given that the areas of life affected by stigma are similar for people with (stigmatized) health conditions [37]. The SSS has not been validated in Amharic. The SSS was translated from English to Amharic, translated back into English again using different interpreters, and pilot tested before use. The SSS was pilot tested among 15 participants (who did not participate in the pilot test of the FQoL) before use.

Qualitative interviews, both in-depth interviews and focus group discussions, were conducted pre-intervention to develop the family-based intervention and post-intervention to evaluate the impact of the intervention.

Four facilitators and seven local area health workers received a four-day training on how to implement the intervention. Pre- and post-intervention date were collected by the four facilitators, all facilitators were local health extension workers who spoke both Amharic and Agew (local) language. The interviewers were trained in the three conditions (both clinical and psychosocial aspects) and interviewing techniques prior to data collection. The questionnaires interview guides were pilot tested before data collection commenced, these participants were not included in the final sample. Minor revisions were made based on the pilot interviews. The interviews were conducted either in participants' homes, in a private space near the patient organisation or at the location of the family-based intervention. The in-depth interviews and focus group discussions were audio recorded. A coordinator monitored the entire process.

## Data analysis

The quantitative data were analysed using SPSS version 24. To correct for the large differences in sample size between baseline and the final sessions, we calculated the differences in physical impairment outcomes by comparing the baseline scores of participants with their last measurement. For some participants, their last measurement was the second or third session. For eye, hand, foot and leg impairments, acute attacks and shoe wearing, paired t-tests were performed to check whether the scores were significantly different between the baseline and the (participants') last measurement. Binary logistic regression was performed to see if there was a relationship between gender and age and physical impairment outcomes at final measurement (number of acute attacks, shoe wearing behaviour and hand, eye and foot impairments). Simple descriptive methods were used to generate a demographic profile of the study sample. Welch's unequal variances t-tests were performed to check whether the scores on the three questionnaires were significantly different pre- and post-intervention (p-value < 0.05). Mean overall scores, scores per participant group and where relevant scores per domain were calculated. Qualitative data were analysed using Open Code 4.03 software. The qualitative data were transcribed and coded using open, inductive coding and content analysis. All data were anonymised before data analysis.

## Results

### Demographic information

A total of 312 persons affected were identified to be included in the family-based intervention. Of the 312 persons listed, 275 (88%) persons affected (115 affected by leprosy and 160 affected by LF or podoconiosis) could be located and were invited to participate. All 275 persons affected who were approached were enrolled and attended the first group meeting with a family member.

A little over half (n = 151, 55%) of the persons affected who participated in the intervention were female. The mean age of the participants was 51 (± 15 SD). An overview of all

**Table 1. Demographic information of the persons affected (n = 275) who attended the first family-based intervention session.**

| | Persons affected by leprosy (n = 115) | Persons affected by lymphatic filariasis or podoconiosis (n = 160) | Total (n = 275) |
|---|---|---|---|
| Average age, mean ± SD | 59 ± 12.7 | 45 ± 14.1 | 51 ± 15.1 |
| Gender, *n* (%) | | | |
| Female | 35 (30.4) | 116 (72.5) | 151 (54.9) |
| Male | 80 (69.6) | 44 (27.5) | 124 (45.1) |
| Living area, *n* (%) | | | |
| Zigem | 5 (4.3) | 160 (100.0) | 165 (60.0) |
| Addis Kidam | 86 (74.8) | 0 (0.0) | 86 (31.3) |
| Injibara | 24 (20.9) | 0 (0.0) | 24 (8.7) |
| Impairments at enrolment*, *n* (%) | | | |
| Eyes | 41 (35.7) | n/a | 41 (14.9) |
| Hands | 42 (36.5) | n/a | 42 (15.3) |
| Feet | 46 (40.0) | 19 (11.9) | 65 (23.6) |
| Legs | n/a | 67 (41.9) | 67 (24.4) |
| Acute attacks | n/a | 137 (85.6) | 137 (49.8) |
| Number of sessions attended, *n* (%) | | | |
| At least one | 115 (100.0) | 160 (100.0) | 275 (100.0) |
| At least two | 91 (79.1) | 157 (98.1) | 248 (90.2) |
| At least three | 73 (63.5) | 142 (88.8) | 215 (78.2) |
| At least four | 63 (54.8) | 102 (63.8) | 165 (60.0) |
| At least five | 51 (44.3) | 23 (14.4) | 74 (26.9) |
| At least six | 37 (32.3) | - | 37 (13.5) |
| At least seven | 18 (15.7) | - | 18 (6.5) |
| At least eight | 8 (7.0) | - | 8 (2.9) |

* Impairments either on the left side, right side or both sides (for example impairments in the left eye, right eye or both eyes). Participants who were lost to follow-up are also included.

demographic information can be found in Table 1. No record was kept of the relationship of family members with the persons affected, however, all family members were household members (parent, child, sibling, or partner).

Table 2 provides an overview of the number of participants out of the total of 275 participants who were invited to participate in the study, who were administered the questionnaires and who were interviewed pre- and post-intervention. Out of the 275 participants who were invited to participate in the intervention, a total of 212 participants (95 persons affected and 117 family members) were administered the FQoL, 94 persons affected the SSS and 71 persons affected the SALSA scale at baseline. Follow-up data was collected of 219 participants for the FQoL (141 persons affected and 78 family members), 149 persons affected the SSS and 126 persons affected the SALSA. We did not keep a record of if participants were interviewed both at baseline and follow-up. In addition, in-depth follow-up interviews were conducted with 25 participants (18 persons affected and 7 family members) and 58 participants were included in a total of nine focus group discussions (40 persons affected and 18 family members).

## Group meeting attendance

There were monthly follow-up visits (group meetings) of the family-based intervention. A total of 74 different group meetings were organised, spread over eight 'sessions' (each session had on average 9 different group meetings). These groups met on different days. On average 12 persons affected with one family member each participated in each meeting.

**Table 2. The total number of participants included per participant group, pre- and post-intervention.**

| | Persons affected by leprosy | | Persons affected by lymphatic filariasis or podoconiosis | | Family members | |
|---|---|---|---|---|---|---|
| | **Baseline** | **Follow-up** | **Baseline** | **Follow-up** | **Baseline** | **Follow-up** |
| FQoL scale | 48 | 73 | 47 | 68 | 117 | 78 |
| SALSA scale | 43 | 75 | 28 | 51 | - | - |
| SARI scale (SSS) | 62 | 78 | 32 | 71 | - | - |
| Interviews | - | 11 | - | 7 | - | 7 |
| Focus group discussions | - | 26 | - | 14 | - | 18 |

Sometimes participants were for example present at the first, third and fourth session, but not at the second. A total of 275 (100%) persons affected attended at least one session, 248 (90%) attended at least two sessions and 215 (78%) attended at least three sessions (Table 1). In most cases, different sessions were held for persons affected by leprosy and their family members, and for persons affected by podoconiosis or LF and their family members, because of the distribution of diseases per district (Table 1).

## Physical outcomes

Table 3 shows the number of persons affected by leprosy who had eye, hand or foot impairments at baseline and at their last follow-up. For a quarter of the participants, their last follow-up was the second or third session (n = 25, 27.5%). There was no change in the number of participants with eye impairments. In addition, logistic regression showed that as age increased, the odds of having eye impairments also increased (p<0.05, odds ratio 1.047, with 95%CI 1.006–1.091). The number of participants with hand impairments decreased at follow-up, but this was not significant. The number of persons affected by leprosy with foot impairments significantly decreased (Table 3). There was no relationship between age and gender and number of acute attacks, shoe wearing behaviour, and hand and foot impairments and between gender and eye impairments at final assessment (p>0.05, logistic regression).

The number of persons affected by LF and podoconiosis who had leg impairments and who had at least one acute attack per month significantly decreased and who wore shoes

**Table 3. The number of persons affected by leprosy with eye, hand and/or foot impairments and the number of participants affected by lymphatic filariasis or podoconiosis with leg impairments, at least one acute attack per month and who wears shoes and at baseline and at participants' last family-based intervention session.**

| | Total number of participants baseline and follow-up | Baseline, $n$ (%) | Final/last assessment[a], $n$ (%) | Difference (%) | p-value[b] |
|---|---|---|---|---|---|
| Has eye impairments | 91 | 31 (34.1) | 31 (34.1) | 0 (0) | NS |
| Has hand impairments | 91 | 36 (39.6) | 33 (36.3) | 3 (8.3) | NS |
| Has foot impairments | 91 | 44 (48.4) | 36 (39.6) | 8 (18.2) | 0.011 |
| Has leg impairments | 145 | 62 (42.8) | 30 (20.7) | 32 (51.6) | 0.000 |
| Has at least one acute attack per month | 146 | 126 (85.7) | 22 (15.2) | 104 (82.5) | 0.000 |
| Wears shoes | 146 | 136 (93.2) | 142 (97.3) | 6 (4.4) | 0.014 |

[a] The final assessment is the last session the participant attended. For persons affected by leprosy this was the second (13% of the participants), third (14%), fourth (9%), fifth (8%), sixth (18%), seventh (13%) or eighth (25%) session, not including the 24 participants that only attended the baseline session. For persons affected by lymphatic filariasis or podoconiosis this was at the second (9%), third (35%), fourth (41%) or fifth (15%) session, the 13 participants that only attended the baseline session not included.

[b] Test used is paired t-test with a significance level of 0.05. NS = not significant (p>0.05).

**Table 4. Leg circumference for persons affected by lymphatic filariasis and podoconiosis at each family-based intervention session.**

| | Baseline | | Final/last assessment[a] | | Difference (%) | 95% CI[b] | p-value |
|---|---|---|---|---|---|---|---|
| | n | Mean (SE) | n | Mean (SE) | | | |
| Circumference right leg | 139 | 26.5 (0.35) | 139 | 24.7 (0.32) | 1.8 (6.8) | 1.4–2.2 | 0.000 |
| Circumference left leg | 132 | 26.8 (0.32) | 132 | 25.1 (0.32) | 1.7 (6.3) | 1.4–2.0 | 0.000 |
| Circumference right foot | 143 | 25.2 (0.22) | 143 | 23.4 (0.21) | 1.8 (7.1) | 1.6–2.2 | 0.000 |
| Circumference left foot | 136 | 25.6 (0.26) | 136 | 23.7 (0.21) | 1.9 (7.4) | 1.6–2.3 | 0.000 |

[a] The final assessment is the last session the participant attended. This was either at the second (for 9–10% of participants), third (36–37% of participants), fourth (40–41% of participants) or fifth (14% of participants) session that was held. The 13 participants that only attended the baseline session are not included.

[b] 95% Confidence interval for mean reduction. Test used is a paired t-test with a significance level of 0.05.

significantly increased between baseline and participants' last follow-up (Table 3). In addition, the mean values of the measurements of the right and left leg (leg *lymphedema*) and right and left foot all significantly decreased between baseline and the final assessment (Table 4).

## Awareness raising

Participants were asked about the awareness raising component of the intervention during the in-depth interviews and focus group discussions. All participants said they were positive about (participating in) the family-based intervention and explained that the intervention had improved their knowledge about the three conditions and of self-management. One participant explained:

> "...Previously I did not think this system [self-care] gives me relief from my illness but after starting implementation of self-care practices, I saw change within two weeks. During the monthly follow-up time research assistants measured my leg circumference and they told me my swelling shows decrement, during this time I felt empowered..." (65-year old man affected by podoconiosis, focus group discussion).

One family member said:

> "...I understood that these diseases are non-communicable after treatment, this gave me high motivation to support my [affected] father..." (20-year old female family member of person affected by leprosy, in-depth interview).

Flyers with disease specific information were also prepared, but only disseminated to the participants after the intervention was completed.

## Socio-economic empowerment

Socio-economic empowerment consisted the formation of establishing Disabled People's Organisations (DPOs). Two DPOs existed already (initiated by the Ethiopian National Association of Persons Affected by Leprosy/ENAPAL, one of the partners in the project): the leprosy specific group in Addis Kidam and in Injibara. A DPO for all three conditions was established in Zigem. Each DPO collected a small contribution fee from its participants, 5 to 20 birr each month (less than one dollar or euro). These fees were used to provide loans for the participants (micro-finance). DPOs also lobbied for 'benefits', e.g. the use of land, from the government. These groups met monthly. While the facilitators of the project helped to establish these

groups and were present during the meeting, they did not give any guidance on the management of the groups. This was done by persons affected themselves.

Both persons affected and their family members said the establishment of the associations was important to them. Participants explained that the association addressed their economic difficulties by providing (self-saved) money:

> "...[The association] encourages saving. We save and can take a loan and use it when there is a challenge..." (65-year old man affected by leprosy, FGD).

Some participants were critical towards the leadership of the (already existing) leprosy association, who they felt not always provided loans to everyone in the organisation.

### Family quality of life, stigma and activity limitations

**Family quality of life.** Table 5 shows the differences in mean family quality of life scores of persons affected and their family members at baseline and follow-up. Higher values indicate better family quality of life. The increase in family quality of life scores for persons affected and for family members are significant (Table 5), indicating that family quality of life has significantly improved after the intervention. Mean family quality of life scores of subgroups improved also, but this difference was not significant for the subgroup of persons affected by leprosy.

When looking at the five domains of the Beach Centre FQoL scale (family interaction, parenting, emotional wellbeing, physical wellbeing and disability-related support), there was a significant improvement in the domains emotional, physical and disability-related support for persons affected at baseline compared to follow-up (p<0.001, unequal variances t-test). The mean scores on these domains have improved over 17%. All five domains have significantly

**Table 5. Mean differences between baseline and follow-up in family quality of life (Beach Centre FQoL), stigma (SARI stigma scale) and activity limitations (SALSA scale) per subgroup.**

| | | Baseline total score, mean (95%CI) | Follow up total score, mean (95%CI) | Difference (%) | p-value[a] |
|---|---|---|---|---|---|
| Family quality of life | All persons affected (n = 95 ~ n = 141) | 76.9 (74.2–79.6) | 84.1 (81.2–87.0) | 7.2 (9.4) | 0.000 |
| | Persons affected leprosy[b] (n = 48 ~ n = 73) | 79.1 (75.2–83.0) | 79.9 (74.4–85.4) | 0.8 (1.0) | NS |
| | People affected podoconiosis or lymphatic filariasis[b] (n = 47 ~ n = 68) | 74.6 (70.9–78.4) | 88.6 (87.8–89.5) | 14 (18.8) | 0.000 |
| | All family members (n = 117 ~ 78) | 67.4 (65.1–69.7) | 89.9 (87.2–92.6) | 22.5 (33.4) | 0.000 |
| Stigma | People affected (total) (n = 94 ~ n = 149) | 24.0 (21.6–26.5) | 16.7 (14.6–18.9) | 7.3 (30.4) | 0.000 |
| | People affected leprosy[b] (n = 62 ~ n = 78) | 22.7 (19.5–26.0) | 16.5 (13.0–20.0) | 6.2 (27.3) | 0.011 |
| | People affected podoconiosis or lymphatic filariasis[b] (n = 32 ~ n = 71) | 26.6 (23.2–30.0) | 17.0 (14.3–19.6) | 9.6 (36.1) | 0.000 |
| Activity limitations | People affected (total) (n = 71 ~ n = 126) | 38.5 (35.1–41.9) | 36.0 (33.9–38.2) | 2.5 (6.5) | NS |
| | People affected leprosy[b] (n = 43 ~ n = 75) | 44.2 (39.8–48.6) | 39.9 (36.7–43.0) | 4.3 (9.7) | NS |
| | People affected podoconiosis or lymphatic filariasis[b] (n = 28 ~ n = 51) | 29.8 (26.2–33.4) | 30.3 (28.6–32.0) | 0.5 (1.7) | NS |

[a] Difference between baseline and follow-up scores, calculated using Welch's unequal variances t-test. NS = not significant (p>0.05).

[b] These are subgroups of the 'persons affected (total)' group.

improved for family members at follow-up (p<0.001, unequal variances t-test). The domains with the biggest mean improvement (>40%) for family members are emotional, physical and disability-related support. An overview can be found in supporting information file S2 Text.

**Stigma.**    Mean (SARI) stigma scores for persons affected significantly decreased from 24.0 at baseline to 16.7 at follow-up. This indicates that stigma for persons affected significantly decreased after the intervention (lower scores on the SARI stigma scale indicate less stigma). A significant decrease was also found for both subgroups (Table 4). Mean scores on three out of the four domains of the SARI stigma scale significantly decreased after the intervention (please see S2 Text), these include experienced, internalised and anticipated stigma (p<0.05, unequal variances t-test). The mean difference on the domain disclosure decreased, but this difference was not significant (p<0.05, unequal variances t-test).

Some participants in the in-depth interviews emphasized that they felt more confident after the intervention. In addition, some participants said their family dynamics and social participation had improved. One participant explained:

*". . .I saw many changes, my wound healed, I had a bad smell before but now after you [project staff] came here my wound is healed. Now I can go to church, I bake injera [Ethiopian flatbread] and prepare a soup like I used to. There is a big change [after the intervention]. . ."* (66-year old woman affected by leprosy, FGD).

Another participant said:

*". . .Previously people discriminated us [the family], we were not allowed to drink coffee with them, but now there is no stigma and discrimination. . ."* (29-year old woman affected by podoconiosis, in-depth interviews).

**Activity limitations.**    With a mean age of 53 (SD 14.4), participants who were administered the SALSA at follow-up were on average a few years older than participants included at baseline (mean 48, SD 16.5). Table 5 shows differences in mean activity limitation scores of persons affected at baseline and follow-up. Lower activity limitations (SALSA) scores indicate less activity limitations. The mean activity limitations decreased at follow-up for persons affected in general and for the persons affected by leprosy subgroup, but increased for the persons affected by podoconiosis and LF subgroup. These differences were not significant (p>0.05, Table 5).

An overview of the different categories of the SALSA (no, mild, moderate, severe or extreme activity limitations) and the number of participants in the in each category at baseline and follow-up can be found as supporting information file (S2 Text). The change in the severe limitations group was the only significant difference between baseline and follow-up. The percentage of persons affected with severe limitations significantly decreased at follow-up (p<0.05, unequal variances t-test, pooled data of persons affected by leprosy, podoconiosis and LF).

## Discussion

Findings from the present study show that the family-based intervention had a positive impact on impairments and self-management of disabilities. Persons affected and family members were enthusiastic and had a positive attitude towards participating in the intervention. In addition, family quality of life improved and stigma decreased at follow-up. Several studies have

found teaching persons affected basic self-care techniques to be a successful approach for morbidity management [15,16,18–20,22,38–41,42], this finding is supported by the present study.

## Family-based format

A new aspects of our intervention is its family-based format. Group-based disability care, often in the form of self-care groups, are common, especially for persons affected by leprosy and LF [15,16,18–20,22,38–41]. To our knowledge, the current intervention is the first family-based intervention for leprosy and podoconiosis-related disability management and the first family-based self-care intervention for leprosy, LF and podoconiosis. Only a few home-based self-care interventions have been conducted for LF [43–45]. In line with our findings, these studies reported good physical impairment outcomes [43–45]. This suggests that family-based self-care is a feasible alternative for self-care groups. Challenges often reported for self-care groups include long distances that need to be travelled to attend group meetings [19], lack of time to attend meetings [46] and sustainability of the groups [18,47]. These challenges do not apply to family-based interventions, where participants do not need to travel to attend and no facilitator is needed. In addition, a family-based intervention is relatively inexpensive since this can be practiced at home and no travel is required. Furthermore, encouragement by significant others can increase motivation, which is essential for self-care behaviour [23,47]. Family-based support therefore seems like a sustainable option, especially in contexts where resources are scarce, which is often the case in areas where leprosy, LF and podoconiosis are endemic [17,19,48–50].

## Components: Self-management of disabilities, awareness, socio-economic empowerment

The intervention in the present study consisted of three components: self-management of disabilities awareness raising, and socio-economic empowerment, with self-management as main component. These three components were developed based on findings from an exploratory study, where we found that many problems faced by participants were not only related to physical impairments, but also caused by stigma and poverty [13]. These findings are supported by a study among over 1,000 persons affected by leprosy in Indonesia, in which the authors stress that "stigma reduction activities and socio-economic rehabilitation are urgently needed in addition to strategies to reduce the development of further physical impairment after release from treatment" [7].

While we have identified 'awareness raising' as a separate component, health education and self-care training are often integrated in self-care and morbidity management training in other studies [16,40,43,44,51]. In addition to providing education about self-care and clinical manifestations during the group sessions, in the present study it was initially also planned to distribute printed material in the communities, to raise awareness and reduce stigma. Due to time constraints, the printed materials were only distributed after the follow-up assessments had been conducted. In leprosy, stigma and a lack of knowledge have been identified as obstacles to case finding and adherence to treatment [52,53]. In addition, social stigma has been associated with poor psychosocial health outcomes [54,55]. Stigma reduction is therefore a crucial component of morbidity management interventions [7]. We recommend including pre- and post-intervention assessment of knowledge and community stigma in future studies, to get a better understanding of the changes in knowledge and community stigma.

Socio-economic empowerment was one of the three components of the intervention in the present study and consisted of the exploration and formation of (disease-specific) Disabled People's Organisations. These organisations/groups collected monthly fees from participants,

that were used to provide loans to group members. Unfortunately, we did not collect any information on how much money was collected by the DPOs as monthly fees, and if any loans were provided. Several studies have emphasized the marginalized position of persons affected in their communities, poverty and a lack of resources for income generation are challenges often reported for persons affected by leprosy, LF and podoconiosis [5,17,19,48–50]. Sometimes, income is too low to acquire basic materials for self-care [46,48]. Costs for treatment, associated non-medical costs and reduced ability to work may cause a financial burden on the entire household of persons affected [5,17,49,50]. We therefore consider socio-economic empowerment an essential component of interventions for self-management and prevention of disabilities. Without income, self-care items such as Vaseline and shoes cannot be bought. Previous studies have also reported positive results in terms of reducing community stigma among persons affected by leprosy, using micro-credit loans and vocational training [56,57]. In addition, if material wellbeing (family income) is positively influenced, family quality of life is higher and all family members, including the affected person, benefit [58].

## Primary outcomes: Physical impairments

In the present study, foot and leg impairments, number of acute attacks, lymphedema and shoe wearing all significantly improved at follow-up. Eye and hand impairments did not improve after the intervention. We believe the lack of improvement in eye impairments relates to their more chronic, permanent nature. Vision loss is not (naturally) reversible. Unfortunately, we did not record severity of impairments. We believe some of the hand impairments reported were either permanent (e.g. loss of digits), may only have been reversed after a longer period of time, or with reconstructive surgery. This is supported by findings from a study in China, that found that regularity in self-care of persons affected by leprosy was only established after three monthly reinforcements [59]. Findings from the present study also indicate that more attention should be paid to self-care practices of hand impairments, something that should be taken into account in future studies.

## Secondary outcomes: Family quality of life, stigma, activities

The intervention in the present study improved family quality of life for persons affected and their family members and decreased stigma. This is an important finding, since leprosy, LF and podoconiosis can negatively impact individual [60–66] and family quality of life [13]. Stigma and visible impairments can also deteriorate quality of life [64]. Quality of life is crucial in the evaluation of health care interventions [67]. Leprosy, LF and podoconiosis-related disabilities require life-long care. Effective morbidity management can lessen impairments and disabilities and is therefore imperative to improve individual quality of life of persons affected [17,67]. It is therefore promising that the intervention in the present study improved family quality of life. We believe the improvement in physical impairments of persons affected in the present study contributed to the improvement in quality of life and reduction in stigma we found, a finding that is supported by a study in Bangladesh [64]. Visible impairments are major determinants of stigma and participation restrictions [7–9].

Even though most physical impairment outcomes improved at the final assessment in the present study, activity levels (assessed by the SALSA scale) did not significantly improve. This is likely related to the lack of improvement found in hand impairments and in part related to the slightly older age (51 years) of the participants in the present study. In addition, during the study we discovered that the SALSA scale is not the most suitable tool for persons affected by LF and podoconiosis. Fifteen out of the total of 20 questions on the SALSA scale relate to hand movements and strength. It is therefore not surprising that persons affected by podoconiosis

and LF had less activity limitations on the SALSA scale, given that these conditions mostly cause lymphoedema of the leg(s) or swelling of other organs like the scrotum [6]. Since the SALSA scale is more sensitive to limitations caused by leprosy, future studies that aim to assess activity limitations among persons affected by LF and podoconiosis would benefit from using a different tool than the SALSA scale.

## Strengths, limitations and recommendations

The present study assessed the short-term outputs and effect of a family-based self-care intervention among a relatively small sample of persons affected by leprosy, LF and podoconiosis. Because of the non-random sample and relatively short follow-up time, effectiveness of the intervention could not be assessed. In addition, the present study did not use a control group of persons not participating in the family-based intervention. A comparison with a control group would have provided additional evidence to the study. However, since this was a proof of concept study, we believe we have demonstrated the feasibility and potential of the family-based intervention. A strength of this study is the mixed-method approach that allowed for triangulation of the data.

To date most prevention of disability effectiveness studies have been flawed by failing to use a randomised controlled design. This has resulted in a lack of evidence about the effectiveness of these interventions. Further research using a randomised controlled design with a larger sample is needed. As a result of this proof of concept study, a randomised controlled trial study using validated measurement tools and more physical impairment outcomes is in development.

Because family structure, roles and functioning are influenced by culture, which in turn influences self-care behaviours, it would be interesting for future studies to explore the self-care behaviours of families in different cultures.

## Conclusions

This proof of concept study showed that the family-based intervention had a positive effect on impairments and self-management of disabilities, family quality of life and stigma. We recommend a large-scale efficacy trial, using a randomised controlled trial and validated measurement tools, to determine its effectiveness and long-term sustainability. Future studies who aim to assess activity limitations among persons affected by lymphatic filariasis and podoconiosis are recommended not to use the SALSA scale, as this scale is more sensitive to limitations caused by leprosy.

## Supporting information

**S1 STROBE Checklist.**
(DOC)

**S1 Text. Description of family-based intervention.**
(DOCX)

**S2 Text. Domain scores for the FQoL, SSS and SALSA.**
(DOCX)

## Acknowledgments

We would like to thank all participants. We are grateful to the research assistants who collected the data: Debritu Bahiru, Addisie Dagnew, Yohannes Woregna and Kassahun Bekele. We

would like to thank the National Podoconiosis Action Network (NaPAN) Ethiopia and the International Orthodox Christian Charities (IOCC) Ethiopia for their support. We want to thank Ms. Femke Boelsma, who contributed to the development of the intervention. We would like to thank Dr Wim van Brakel from NLR for his technical advice. In addition, we would like to thank Prof Jan Hendrik Richardus of Erasmus MC, University Medical Center for his careful reading of the final draft.

## Author Contributions

**Conceptualization:** Anna T. van't Noordende.

**Data curation:** Moges Wubie Aycheh.

**Formal analysis:** Anna T. van't Noordende.

**Funding acquisition:** Anna T. van't Noordende, Moges Wubie Aycheh, Tesfaye Tadesse, Tanny Hagens, Alice P. Schippers.

**Investigation:** Moges Wubie Aycheh.

**Methodology:** Anna T. van't Noordende.

**Project administration:** Tesfaye Tadesse, Tanny Hagens, Eva Haverkort.

**Resources:** Tesfaye Tadesse, Tanny Hagens.

**Software:** Anna T. van't Noordende, Eva Haverkort.

**Supervision:** Tesfaye Tadesse, Alice P. Schippers.

**Validation:** Anna T. van't Noordende, Moges Wubie Aycheh.

**Visualization:** Anna T. van't Noordende.

**Writing – original draft:** Anna T. van't Noordende.

**Writing – review & editing:** Anna T. van't Noordende, Moges Wubie Aycheh, Tesfaye Tadesse, Tanny Hagens, Eva Haverkort, Alice P. Schippers.

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
