## [Decision Letter · Decision Letter 0]

23 Oct 2020

Dear Dr. Noordend,

Thank you very much for submitting your manuscript "A family-based intervention for prevention and self-management of disabilities due to leprosy, podoconiosis and lymphatic filariasis in Ethiopia: A proof of concept study" for consideration at PLOS Neglected Tropical Diseases. As with all papers reviewed by the journal, your manuscript was reviewed by members of the editorial board and by several independent reviewers. In light of the reviews (below this email), we would like to invite the resubmission of a significantly-revised version that takes into account the reviewers' comments. 

We cannot make any decision about publication until we have seen the revised manuscript and your response to the reviewers' comments. Your revised manuscript is also likely to be sent to reviewers for further evaluation.

Sincerely,

Jessica K Fairley, MD, MPH

Associate Editor

Gerson Penna

Deputy Editor

Reviewer's Responses to Questions

**Key Review Criteria Required for Acceptance?**

**Methods**

-Are the objectives of the study clearly articulated with a clear testable hypothesis stated?

-Is the study design appropriate to address the stated objectives?

-Is the population clearly described and appropriate for the hypothesis being tested?

-Is the sample size sufficient to ensure adequate power to address the hypothesis being tested?

-Were correct statistical analysis used to support conclusions?

-Are there concerns about ethical or regulatory requirements being met?

Reviewer #1: This study is well designed, objectives al clear and well adreessed throughout the text. The population is well described and the statistical analysis is adequate for the purpose of the study. The text reflects the existence of the necessary documents to comply with ethical and regulatory requirements.

Reviewer #2: Objectives - this is a 'proof of concept' study, so the design is appropriate (and the limitations have been clearly listed.)

Design - is okay as a proof of concept - but for proper testing of effectivenesss a RCT is needed (aithors have suggested this in the conclusions).

Reviewer #3: - The objective of the study is clearly stated and the study design selected is appropriate for objective.

- The study population is appropriate for the objective

- There is a difference in the sample size determined and the total sample included in the study. Is the sample size calculated for family members or affected individual? 

- Line 149-153: the calculated sample size is 30 families affected by leprosy and 30 families of persons affected by LF or Podoconiosis. But the total sample included is larger than calculated. Why?

- Line 172-173: The sampling strategy to include people affected by leprosy appears to be flawed. Why didn’t they select participants (for people affected by leprosy) from lists available in the health post just like the strategy used to include LF/Podoconiosis (line 169-172).

- The statistical tests used were appropriate for the study design (pre/post design), hence supports the conclusion.

- Some concerns: The awareness raising intervention was not implemented as planned (line 351-352, 487-488) …..the printed materials were only distributed after the follow-up assessments had been conducted. This raises the question on attribution of awareness raising on outcome. Line 337-338 claim on awareness raising appears to be due to education give as part of self-management than awareness raising. 

- Line 253-254: this is the intervention study, written consent would have been ideal.

**Results**

-Does the analysis presented match the analysis plan?

-Are the results clearly and completely presented?

-Are the figures (Tables, Images) of sufficient quality for clarity?

Reviewer #1: The analysis presented match the plan. Results are clearly and completely presented and tables are of sufficient quality

Reviewer #2: One of the consequences of convenience sampling is that your two populations are not comparable with persons affected by leprosy being much older than those affected by LF and podo, so the increased activity limitation is to be expected.

Line 275 - typo - should be 'out of the 312 participants...'

Line 278 - typo - '147' should be '179' (cf tabel 2)

Line 320 - The text saus that the number who wore shoes decreaed - but in table 3 the number of people who wore shees increased (from 126 to 142)

Fig 1 - the Y axis will be clearer if the score range for each of the categories was included No = 10-24; Mild = 25-39; Mpderate = 40-49; Severe 50-59; Extreme 60-80. (See https://www.leprosy-information.org/media/627/download)

Line 518/519 - some impairments are permanent - amputations, loss of digits, fixed contractures etc.

Reviewer #3: - Overall, the analysis plan matches the results presented. Except, for linear regression that the author intended to do is not reported in the results. 

- Line 258-262: When the calculated sample size is 30 family members each among people affected by leprosy and LF/Podoconiosis, why would the investigator included 275 participants. Any justification?

- Line 275-282: the description of table 2 is not clear. 

- Line 292-294: shows number of people in the follow-up has come down. Table 2 shows increase in number during follow-up?

- Table 2: What is the justification for difference in number of participants interviewed for questionnaire between baseline and follow-up.

- Line 183-184: Simplified EHF score was measured to study the physical outcome, but not reported. Is Table 3 relating to EHF score? 

- Please provide reference for simplified EHF score

- line 303-304: the reported relationship between age with post intervention physical outcome is not clear. The 95CIs should be given for the odds ratio.

- Line 320: people who wore shoes increased among people with LF and Podoconiosis

- Change in the shoe wearing among people affected by leprosy is not reported.

- Line 323-325: is the comparison between men and women for baseline measurement or at final follow-up, not clear.

- Table 4: Standard deviation or Standard Error to be provided for mean. If relevant, Median and IQR value can be provided. 

- Line 355-364: DPOs for people affected by leprosy already existed. No data on socio-economic intervention provided through this research is provided to justify its effect on the outcome.

- Line 392-398 and 401-407: Table not provided for the results explained here.

- Line 436-437: is significant change in severe category for pooled data or for people affected by leprosy? 

- Figure 1: A simple cross-tab comparing different categories of SALSA score for baseline and follow-up would be more useful than seeing the change in the percentage in individual category. 

- The qualitative data analysis not provided

**Conclusions**

-Are the conclusions supported by the data presented?

-Are the limitations of analysis clearly described?

-Do the authors discuss how these data can be helpful to advance our understanding of the topic under study?

-Is public health relevance addressed?

Reviewer #1: The conclusions are fully supported, the limitations are clearly explained and are accurate. The discussion can be written more succinctly to be presented to the reader more clearly. 

Public health relevance is clearly explained as it corresponds to the actual public relevance of the intervention

Reviewer #2: Conclusions are somewhat supported, but a full RCT is recommended which reflects the limit of this study

A number of limitations are acknowledged - hence the need for a RCT

The comment about the limits of the SALSA scale is appreciated.

No reference to the relevance to public health

Reviewer #3: - Line 574-575: given the shorter duration of the intervention and the observation period it would be too ambitious to conclude as ‘impact’ of the intervention. Line 553-554 author says it is the short-term outputs. The observed change is the short-term outcome not impact.

- Another limitation is that the implementation of the intervention and outcome measurements were done by the same staff so possibility of measurement bias.

**Editorial and Data Presentation Modifications?**

Reviewer #1: The study is very interesting and presents a very useful kind of intervention to be implemented in other areas, and its design is adequate. I believe, however, that it can be presented, without losing information, in a more concise way. I believe that it is longer than it can be, and that its wording can be greatly reduced without undermining the ideas, the results and the discussion and reasoning that the authors want to convey to the reader. This is especially recommended in the chapter "Discussion". With regard to table n1, consider reducing the list of "sessions attended" by making groups (1; 2-4; 5-8).

Reviewer #2: Line 33: reorder the the words to 'activity limitations, stigma and family quality of care' to match the order the tools are listed

line 184 - typo remove 'in'

Reviewer #3: - Line 22-23: Not all patients with leprosy and LF will require life-long need to practice self-management… only those with impairment would need life-long care 

- Line 41: disability management.. elsewhere in the manuscript it is mentioned as self-management. Consistent use of term would be very helpful.

- Line 75-55: Ref 16 is for the heart condition

- Line 121-122: not clear

- Line 455, 459: New aspects of our intervention are its integrated approach and ……. meaning of integrated is not clear?

- Line 165: Persons unwilling… is not an exclusion criteria

- Line 183-185: the data on simplified EHF score is not reported

- Line 197, 214: are they same as study participants or different people?

- Line 208: the word “assesses”

- Line 275-282: the description is confusing. Should be simplified.

- Line 304: 95% CIs to be provided for odds ratio

- Line 392-398 & 401-407: data of sub scores to be provided for easy understanding

- Line 502-506: author appears to be indicating that they have increased the income through socio-economic intervention, if so, the data should be added

- Line515-516: the explanation of odds ratio is not clear.

**Summary and General Comments**

Reviewer #1: Same as above: The study is very interesting and presents a very useful kind of intervention to be implemented in other areas, and its design is adequate. I believe, however, that it can be presented, without losing information, in a more concise way. I believe that it is longer than it can be, and that its wording can be greatly reduced without undermining the ideas, the results and the discussion and reasoning that the authors want to convey to the reader. This is especially recommended in the chapter "Discussion". With regard to table n1, consider reducing the list of "sessions attended" by making groups (1; 2-4; 5-8).

Reviewer #2: I am struggling to really comprehend what is unique about FBA. what are the family members contributing to the gorup functioning and dynamics? Apart from the presence of family members, what is different abotu FBA to 'traditional' self-care groups? I think a few more sentences making this clear would strengthen the paper.

I also have a question about sustainability of this 'proof of concept' - when there is a heavy investment of vaseline, bucket, soap and bandages. Does thes supply of these things counfound the results?

Data collection - the simplified EHF - is possibly too simple and not clear. EHF gives us a range of 0-12. Is this simplified on a range of 0-3 or 0-6. That is, are you counting the hands together or seperately. If you are counting them together, then avoid the term EHF score - as that is not what you have done. I see the small note under table 1, but the narrative could benefit from a small rewrite.

The paper ends by telling us that a RCT is in development - this will clear up some of the issues I have listed here.

Reviewer #3: - The sustainable solution for care of people affected by leprosy, LF and podoconiosis is essential to improve the quality of life of person affected and this study piloted the family based approach as a possible solution. 

- The impact of awareness raising and socio-economic intervention on the outcome is doubtful given the fact that awareness rising was not implemented as intended and the study period is too short to attribute role of socio-economic intervention on the outcome.

- EHF score is a known sensitive indicator of change in the impairment status among people affected by leprosy. Author measured the EHF score but did not report.

- The word integrated appears in the discussion. The author should be clear on what they meant by integrated. Are they meaning bringing affected person and family together as integrated or between the disease group?

- The logistic regression and linear regression are mentioned in the analysis plan but not reported.

PLOS authors have the option to publish the peer review history of their article (what does this mean?). If published, this will include your full peer review and any attached files.

Reviewer #1: No

Reviewer #2: No

Reviewer #3: No
---

## [Decision Letter · Decision Letter 1]

20 Jan 2021

Dear Dr van 't Noordende,

We are pleased to inform you that your manuscript 'A family-based intervention for prevention and self-management of disabilities due to leprosy, podoconiosis and lymphatic filariasis in Ethiopia: A proof of concept study' has been provisionally accepted for publication in PLOS Neglected Tropical Diseases.

Best regards,

Jessica K Fairley, MD, MPH

Associate Editor

Gerson Penna

Deputy Editor

Reviewer's Responses to Questions

**Key Review Criteria Required for Acceptance?**

**Methods**

-Are the objectives of the study clearly articulated with a clear testable hypothesis stated?

-Is the study design appropriate to address the stated objectives?

-Is the population clearly described and appropriate for the hypothesis being tested?

-Is the sample size sufficient to ensure adequate power to address the hypothesis being tested?

-Were correct statistical analysis used to support conclusions?

-Are there concerns about ethical or regulatory requirements being met?

Reviewer #1: Objetives and design are clear and appropiate

Reviewer #2: (No Response)

**Results**

-Does the analysis presented match the analysis plan?

-Are the results clearly and completely presented?

-Are the figures (Tables, Images) of sufficient quality for clarity?

Reviewer #1: I've not concerns about results

Reviewer #2: (No Response)

**Conclusions**

-Are the conclusions supported by the data presented?

-Are the limitations of analysis clearly described?

-Do the authors discuss how these data can be helpful to advance our understanding of the topic under study?

-Is public health relevance addressed?

Reviewer #1: Conclusions are suppoted, public health relevance and limitations described

Reviewer #2: (No Response)

**Editorial and Data Presentation Modifications?**

Reviewer #1: (No Response)

Reviewer #2: (No Response)

**Summary and General Comments**

Reviewer #1: No comments after authors modifications

Reviewer #2: (No Response)

PLOS authors have the option to publish the peer review history of their article (what does this mean?). If published, this will include your full peer review and any attached files.

Reviewer #1: No

Reviewer #2: No

---

## [Editor Report · Acceptance letter]

12 Feb 2021

Dear Ms. van 't Noordende,

We are delighted to inform you that your manuscript, "A family-based intervention for prevention and self-management of disabilities due to leprosy, podoconiosis and lymphatic filariasis in Ethiopia: A proof of concept study," has been formally accepted for publication in PLOS Neglected Tropical Diseases.

Best regards,

Shaden Kamhawi

co-Editor-in-Chief

Paul Brindley

co-Editor-in-Chief
